# SCREWS 🔩 : A Modular Framework for Reasoning with Revisions

## Abstract

Large language models (LLMs) can improve their accuracy on various tasks through iteratively refining and revising their output based on feedback. We observe that these *revisions* can introduce errors, in which case it is better to roll back to a previous result. Further, revisions are typically homogeneous: they use the same reasoning method that produced the initial answer, which may not correct errors. To enable exploration in this space, we present SCREWS, a modular framework for reasoning with revisions. It is comprised of three main modules: *Sampling*, *Conditional Resampling*, and *Selection*, each consisting of sub-modules that can be hand-selected per task. We show that SCREWS not only unifies several previous approaches under a common framework, but also reveals several novel strategies for identifying improved reasoning chains. We evaluate our framework with state-of-the-art LLMs (ChatGPT and GPT-4) on a diverse set of reasoning tasks and uncover useful new reasoning strategies for each: arithmetic word problems, multi-hop question answering, and code analysis. Heterogeneous revision strategies prove to be important, as does selection between original and revised candidates.

## 1 Introduction

Large Language Models (LLMs) have proven effective on a variety of reasoning tasks (OpenAI, 2023). However, the LLM output is not always correct on its first attempt, and it is often necessary to iteratively refine the outputs to ensure that the desired goal is achieved (Madaan et al., 2023; Welleck et al., 2022; Zheng et al., 2023). These refinement methods assume that subsequent outputs (either by the same model, or by an external model or some tool) lead to better performance. However, there is no guarantee that subsequent versions must be better; as Fig. 1 illustrates, refinement can lead to a wrong answer. This motivates a *Selection* strategy whereby the model can select an earlier output.

In addition, past work on iterative refinement typically assumes a single, fixed reasoning strategy (Welleck et al., 2022; Huang et al., 2022; Madaan et al., 2023; Zheng et al., 2023). Humans, however, are more flexible. A student preparing for an exam may use deductive reasoning to solve problems and inductive reasoning to verify the results; or a product manager may use a brainstorming strategy to list several ideas and then a prioritization strategy to rank them based on their feasibility or impact. Thus, we propose a *modular* approach to answer refinements, allowing us to test different strategies.

In this work, we introduce SCREWS, a modular framework for reasoning with revisions.[1] Fig. 2 introduces the three main modules of the framework in detail, namely *Sampling*, *Conditional Resampling*, and *Selection*. For a given task and input sequence, we instantiate SCREWS by fixing the submodules for each module (for example, we might select "Chain of Thought" for *Sampling*). The initial outputs generated by *Sampling* are passed to *Conditional Resampling*, which decides whether to generate a revision *conditioned* on the initial sample, and does so if needed. Finally, all samples and revisions are given to the *Selection* module, which selects the best one. Given the modular nature of our framework, several recently proposed self-refining methods can be improved by using other components of the framework. An example is the combination of the self-refinement

---

[1] SCREWS 🔩 stands for "**S**ampling, **C**onditional **RE**sampling **W**ith **S**election." We will release the code.

method (Madaan et al., 2023) with our model-based selection strategy, which can improve overall performance; more such strategies are described in Sec. 5.

We evaluate SCREWS on a variety of reasoning tasks: arithmetic reasoning, multi-hop question answering, and code analysis, using ChatGPT (Brown et al., 2020) or GPT-4 (OpenAI, 2023). Our proposed strategies achieve substantial improvements (10–15%) over vanilla strategies of sampling and resampling. We demonstrate the usefulness of heterogeneous resampling, whereby the model modifies its reasoning, leading to a substantial improvement over the baselines at a very low overall cost. We also discuss the importance of a model-based selection strategy that allows the model to roll back to its previous more confident outputs, an important component for modern LLMs.

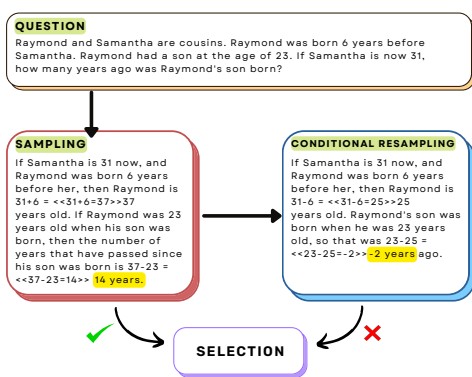

Figure 1: An example demonstrating that *Conditional Resampling* (also known as "*refinement*") can lead to incorrect modification of the original answer. The *Selection* module can retract it, if needed.

## 2 BACKGROUND

**Sampling** Prompting LLMs to generate a series of intermediate steps has proven to be effective for improving their reasoning capabilities (Wei et al., 2022; Lewkowycz et al., 2022; Kojima et al., 2022; Wang et al., 2022). Some approaches in this direction include Chain of Thought (Wei et al., 2022; Zhang et al., 2022; Wang et al., 2022) and adding "Let's think step by step" to the prompt (Kojima et al., 2022). Another approach is "question decomposition", which decomposes the main problem into simpler problems and solves them iteratively (Min et al., 2019; Shridhar et al., 2022; Zhou et al., 2022; Jhamtani et al., 2023; Radhakrishnan et al., 2023). Each of these approaches has its own advantages depending on the underlying task (Shridhar et al., 2023). However, we are not aware of work combining these methods.

**Conditional Resampling** The use of feedback to improve generated samples has been well studied, where the feedback can come either from humans (Tandon et al., 2021; Bai et al., 2022; Elgohary et al., 2021), from reward models (Ziegler et al., 2019; Lu et al., 2022; Shridhar et al., 2022; Christiano et al., 2017; Lightman et al., 2023), from external tools such as code interpreters (Schick et al., 2023; Chen et al., 2022), or from other LLMs (Madaan et al., 2023; Welleck et al., 2022; Fu et al., 2023; Peng et al., 2023; Yang et al., 2022; Zheng et al., 2023; Cohen et al., 2023; Ling et al., 2023; Khalifa et al., 2023). However, even if these feedback mechanisms are infallible, the resulting revisions may introduce new errors. While prior work uses the term "refinement," we do not because refinement implies finer (improved) responses, which is not always the case.

**Selection** When using LLMs revise the output, a common selection technique is to select the final result (Madaan et al., 2023; Shinn et al., 2023; Zheng et al., 2023; Yao et al., 2022; Chen et al., 2023; Weng et al., 2022). However, this can lead to accepting incorrect changes made to previously correct results. Other selection methods involve ranking multiple sampled outputs (Burges et al., 2005; Cobbe et al., 2021) or majority voting (Wang et al., 2022; Lewkowycz et al., 2022; Zheng et al., 2023). These methods often use a homogeneous sampling strategy with changes in hyperparameters. Our work extends the strategy to heterogeneous sampling and selection.

## 3 SCREWS: METHODOLOGY

In this section, we describe SCREWS, our proposed modular framework for reasoning with revisions to tackle different reasoning tasks. Given a problem x, the goal is to generate an *answer a*, which in our experiments may be a string or a number. SCREWS consists of three main modules: *Sampling*, *Conditional Resampling*, and *Selection*. Different variants of SCREWS are obtained by instantiating these modules in different ways. The options for each module are described below and illustrated schematically in Fig. 2. Note that there are other possible ways to instantiate each module. However in this work, we study only the instantiations described below.

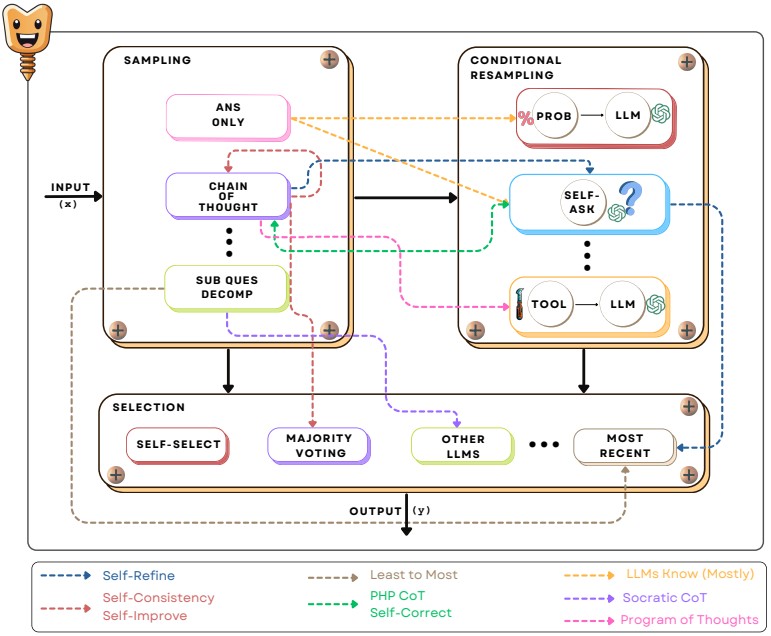

Figure 2: Overview of our modular framework for reasoning with revisions, SCREWS. Each of the three large boxes ("modules") contains several alternatives ("submodules"). Several past works can be viewed as instances of our framework, namely Self-Refine (Madaan et al., 2023), Least to Most (Zhou et al., 2022), LLMs Know (Mostly) (Kadavath et al., 2022), Self-Consistency (Wang et al., 2022), Self-Improve (Huang et al., 2022), PHP CoT (Zheng et al., 2023), Self-Correct (Welleck et al., 2022), Socratic CoT (Shridhar et al., 2022), Program of Thoughts (Chen et al., 2022), among many others. (...) represents other sub-components that can be added to each module, like cached memory or web search for *Sampling*, among others.

All of our methods will invoke one or more stochastic functions, where each function $\psi$ maps a tuple of input strings to a *result* string $\mathrm{y}$ that contains useful information. In practice, $\psi$ deterministically constructs a prompt from the input strings and then samples $\mathrm{y}$ from a large pretrained language model as a stochastic continuation of this prompt. For a given tuple of input strings, the prompt constructed for $\psi$ will typically be a formatted encoding of this tuple, preceded by a task specific instruction and several demonstrations (few-shot examples) that illustrate how $\psi$ should map other encoded input tuples to their corresponding continuations (Brown et al., 2020).

## 3.1 SAMPLING

We consider three instantiations of the *Sampling* module, each may be suitable for different tasks.

**Answer Only** In this method, for a given problem $\mathrm{x}$, the model $\psi$ directly generates the answer $\mathrm{y} = \psi(\mathrm{x})$ without any intermediate steps. This is the simplest and most naive sampling method. The value of $\mathrm{y}$ is returned as the answer $a$ (if there is no further revision of $\mathrm{y}$).

**Chain of Thought (CoT)** For many reasoning tasks today, generating explanations improves the quality of the final answer (Wei et al., 2022; Kojima et al., 2022). Chain of Thought sampling encourages the model to explain the intermediate step-by-step reasoning en route to a decision. This approach is now commonly used in several reasoning tasks. Again, we define $\mathrm{y} = \psi(\mathrm{x})$, but now we expect the prompt continuation to consist of step-by-step reasoning culminating in the step by step answer $\mathrm{y}$, as demonstrated by the few-shot examples included in the prompt. The answer $a$ is extracted from $\mathrm{y}$ using a simple deterministic pattern-matching heuristic.

**Sub-question decomposition** This method decomposes the problem $\mathrm{x}$ into simpler sub-questions $[x_1, x_2, \ldots, x_n]$. For each sub-question $x_i$ in turn ($i = 1, 2, \ldots, n$), the model is called to generate the corresponding sub-answer $y_i = \psi(\mathrm{x}, x_1, y_1, \ldots, x_{i-1}, y_{i-1}, x_i)$. Note that we generate all ques-

tions before seeing any answers; that choice follows Shridhar et al. (2023), who found this approach to work better than interleaved generation of questions and answers. The sequence of questions may be generated in a single step, either by a call to a stochastic function $\psi_{\text{question}}$, or by a custom question generation module that has been fine-tuned on human-written questions as in Cobbe et al. (2021). The answer $a$ is extracted from $y_n$ with a simple heuristic as in CoT.

## 3.2 CONDITIONAL RESAMPLING

The result $y$ from the *Sampling* module can be viewed as a *provisional result*, $y_{\text{curr}}$. This is passed to the *Conditional Resampling* module where a decision is made whether or not to revise it. This is done in two steps: first deciding whether or not to revise, and then if so, resampling a new result $y_{\text{next}}$ using one of the sampling methods mentioned above. The resampling is conditional because $y_{\text{next}}$ may depend on $y_{\text{curr}}$. Our work focuses on the following instantiations for *Conditional Resampling*:

**Self-Ask** Kadavath et al. (2022) uses a function $\psi_{\text{ask}}(x, y_{\text{curr}})$. The first token of the result indicates whether $y_{\text{curr}}$ is correct, for example by starting with "Yes" or "No". If "Yes", we do not resample; if "No", we must resample a revised answer $y_{\text{next}}$. In principle, the revision could be iterated, although Kadavath et al. (2022) did not do this, nor do our experiments in this paper.

In our version of self-ask, $\psi_{\text{ask}}$ is formulated so that $y_{\text{next}}$ appears in the result string $\psi_{\text{ask}}(x, y_{\text{curr}})$ following the token "No". Thus, both steps are efficiently performed by a single call to $\psi_{\text{ask}}(x, y_{\text{curr}})$. For this method, we always use greedy decoding (temperature 0) to deterministically select whichever of "Yes" or "No" is more probable.[2]

When the sampling module (Sec. 3.1) used sub-question decomposition to produce a chain of sub-answers $y_{\text{curr}} = [y_1, \ldots, y_n]$, rather than checking and revising only the final result step $y_n$ by calling $\psi_{\text{ask}}(x, y_n)$, we can instead check and revise each step, at the cost of more calls to $\psi_{\text{ask}}$. For each provisional sub-answer $y_i$ in turn (starting with $i = 1$), we predict whether it is correct by calling $\psi_{\text{ask}}(x, x_1, y_1, \ldots, x_{i-1}, y_{i-1}, x_i, y_i)$. The first time the output is "No", we resample $y'_i$ through $y'_n$, yielding the revised result $y_{\text{next}} = [y_1, \ldots, y_{i-1}, y'_i, \ldots, y'_n]$. In principle, self-ask could then be applied again at later steps $> i$ of both the original and revised chains; then choosing among the many resulting chains, using the selection procedures of the next section, would resemble branching in a reasoning tree (Yao et al., 2023).

**Tool use** For some tasks, we construct $\psi_{\text{ask}}$ so that it is allowed to use tools (Schick et al., 2023). The reason is that in tasks like fact-checking, it is futile to ask the LLM to check $y_{\text{curr}}$ because it might not have the requisite knowledge for evaluation. The tools can be used to collect additional information to help the model detect and fix problems in its own generated answer. Tools like search engines or fact retrievers can be used to evaluate correctness and generate a new revision. Other tools like code interpreters are not capable of generating text, but can still be used to evaluate correctness.

## 3.3 SELECTION

The last module in SCREWS is the *Selection* module. In this step, we use either a model $\psi_{\text{select}}$ or simple heuristics to select the *final* result $y$ from which we then extract the *final* answer $a$. In effect, this allows us to construct a simple ensemble of multiple systems.

**LLM-Based Selection** Just as an LLM was used above to evaluate whether $y_{\text{curr}}$ is good, an LLM can be used to evaluate whether $y_{\text{next}}$ is better. We call $\psi_{\text{select}}(x, y_{\text{curr}}, y_{\text{next}})$ to choose between two result strings.[3] Note that it could be naturally extended to choose among more than two answers. When selection and sampling are implemented using the same LLM, we refer to the method as *self-select* (e.g., in Fig. 2).

**Rule-Based Selection** We consider the other methods we study to be rule-based. Past work on iterative refinement (Madaan et al., 2023; Huang et al., 2022; Zheng et al., 2023) always selects the most recent revision. Majority voting is a simple traditional ensembling method that has been used for selection (Wang et al., 2022; Lewkowycz et al., 2022), but it is costly since it requires several samples.

---

[2]A threshold other than 50% could also be selected. Alternatively, the correctness probability of $y_{\text{curr}}$ could be assessed by a dedicated $\psi_{\text{check}}(x, y_{\text{curr}})$, but we were unsuccessful with this as $\psi_{\text{check}}$ was poorly calibrated, mirroring findings on model calibration (Kadavath et al., 2022; Xiong et al., 2023).

[3]We found the order of $y_{\text{curr}}$ and $y_{\text{next}}$ in the prompt was unimportant, so we randomized it in our study.

# 4 EXPERIMENTS

## 4.1 TASKS

We test the effectiveness and flexibility of SCREWS on three categories of reasoning tasks: GSM8K (Cobbe et al., 2021) for arithmetic reasoning, StrategyQA (Geva et al., 2021) for multi-hop question answering, and Big-Bench (BIG-bench authors, 2023) Auto Debugging[4] for code analysis. GSM8K is a grade-school-level math word problem dataset with a test set of 1319 samples, each requiring two to eight steps to solve. GSM8K includes sub-questions that were generated by a fine-tuned GPT-3 model and correspond to the steps in a particular correct CoT solution. Since these sub-questions were generated with oracle knowledge of a correct CoT solution, we refer to experiments using them as "Subq (Or)". We use "Subq (QG)" for the fairer experimental condition where we instead generated the sub-questions from ChatGPT using 2-shot prompts (shown in Appendix B.4).

Following Magister et al. (2023) and Shridhar et al. (2023), we test on the first 490 samples from the training set of StrategyQA (since their test set is unlabeled). The demonstration examples for our various stochastic functions $\psi$ were drawn randomly from the rest of the training set. StrategyQA also includes human-annotated oracle subquestions (again denoted with "Subq (Or)") and related facts that can assist in answering the main question (which we use for tool-based conditional resampling as in Sec. 3.2). The Auto Debugging dataset tests whether a model can answer questions about the intermediate state of a program without executing the code. The dataset consists of 34 coding examples, of which 33 were used as test examples and 1 as a demonstration example in the prompt.

## 4.2 EXPERIMENTAL SETUP

We always report exact-match accuracy: the percentage of examples on which our final answer $a$ matches the gold answer. For all of our experiments, we use the ChatGPT API (Brown et al., 2020) from July 2023 (`gpt-3.5-turbo-0301`).

**Sampling** With all choices of the *Sampling* module, we use 5-shot sampling for GSM8K and StrategyQA and 1-shot sampling for Auto Debugging (Appendix B.1). Greedy decoding (temp = 0) is used for the main experiments while higher temperature (0.7) is used for the majority voting experiments (one sample was generated greedily and the others at temp = 0.7).

**Conditional Resampling** Greedy decoding is used to first make a binary resampling decision and then to sample. 4-shot prompts (with two correct and two incorrect samples) are used for the GSM8K and StrategyQA datasets, while a 2-shot prompt (with one correct and one incorrect sample) is used for Auto Debugging (Appendix B.2). For StrategyQA, we use tool-based resampling by including the provided facts from the dataset into the prompt to simulate a (perfect) fact retrieval tool.

**Selection** For the *self-select* strategy, the prompts (Appendix B.3) include two examples and selection was produced with greedy decoding. For majority voting, a majority vote on the final answers was taken over $k \in \{1, 3, 4, 5\}$ samples. Ties were broken randomly.

# 5 RESULTS

## 5.1 GSM8K

**Conditional Resampling Works Better with Method Change**  Previous work (Madaan et al., 2023) has shown that when a chain-of-thought method is used for initial *Sampling*, reasoning ability is improved by *Resampling* with the same method, taking the previous sample into account.

We reproduced this previous finding: the CoT scores for GSM8K improved by 1.4 points after resampling with CoT (71.6 to 73.0), as shown in Tab. 1. However, when the initial *Sampling* used subquestion decomposition, we found that resampling with subquestion decomposition actually harmed accuracy (71.9 to 71.3 with Subq (QG), 78.6 to 78.2 with Subq (Or)).

What gave the best results—for all three *Sampling* methods—was *Conditional Resampling* with a *different* method from the originally chosen one. It gave a large gain over Sampling when the

---

[4] https://github.com/google/BIG-bench/tree/main/bigbench/benchmark_tasks/auto_debugging/

Table 1: The improvements achieved by using *Conditional Resampling* for the GSM8K dataset, where $y_{next}$ is **always** selected (* indicates statistical significance with $p < 0.05$). Sec. 4.1 describes CoT, Subq (QG), and Subq (Or) in more detail.

| Sampling | Accuracy | Conditional Resampling | Accuracy |
|---|---|---|---|
| **CoT**: Chain of thought | 71.64 | CoT | 73.00* |
| | | Subq (QG) | 73.69* |
| | | Subq (Or) | **73.99**\* |
| **Subq (QG)**: Subquestion decomposition with ChatGPT-generated questions | 71.87 | CoT | **73.99**\* |
| | | Subq (QG) | 71.26 |
| | | Subq (Or) | 72.80 |
| **Subq (Or)**: Subquestion decomposition with oracle questions present in GSM8K | 78.62 | CoT | **78.99** |
| | | Subq (QG) | 78.86 |
| | | Subq (Or) | 78.24 |

original Sampling used CoT and Resampling used subquestion decomposition (71.6 to 73.7, with generated subquestions) and vice versa (71.9 to 74.0). Even with oracle subquestions, moderate gains are still seen when resampling with CoT (78.6 to 79.0). This demonstrates that it is useful to change methods using *Conditional Resampling*, a novel finding with our framework.

Table 2: Impact of *Selection* on the GSM8K data set on *Independent Sampling* and *Conditional Resampling*. The upper bound from using a *Selection* oracle is given in square brackets.

| Method | Independent Sampling | | | Conditional Resampling | | |
|---|---|---|---|---|---|---|
| | CoT | Subq (QG) | Subq (Or) | CoT | Subq (QG) | Subq (Or) |
| CoT | 71.64 | 74.90 [85.36] | **81.34** [89.08] | 72.93 [73.08] | 73.76 [73.76] | **73.99** [73.99] |
| Subq (QG) | 74.90 [85.36] | 71.87 | **79.99** [87.26] | **73.99** [75.43] | 72.40 [72.40] | 73.84 [76.04] |
| Subq (Or) | **81.34** [89.08] | 79.99 [87.26] | 78.62 | 78.99 [81.50] | **79.75** [80.97] | 79.22 [79.22] |

**Importance of Selection Module** *Conditional Resampling* does not invariably improve every output. In fact, we saw in Tab. 1 that for some settings, it may harm the output quality even on average. This is why the *Selection* module is useful—to detect and reject cases of harmful revisions.

First, as a starting point, the left half of Tab. 2 considers using Selection only as an ensembling technique to combine the outputs of two *independent* Sampling strategies. (Note that this matrix is symmetric.) Although CoT and subquestion decomposition are about equally good Sampling strategies (71.6 and 71.9), using a Selection module to select the better of the two achieves a 3-point gain (to 74.9). Much larger gains (up to 85.4) are potentially available from improving Selection—the upper bound on performance (if Selection always chose the better option) is shown in square brackets. This shows that the two Sampling strategies have largely complementary errors. A similar pattern applies when the subquestion decomposition method is permitted to use oracle subquestions, which improves performance across the board to 81.34.

The right half of Tab. 2 shows *Selection* between the *Sampled* and *Conditionally Resampled* predictions from Tab. 1. (This matrix is asymmetric.) For CoT, the results remain the same at 73.99, which is due to the fact that the upper bound is at 73.99, showing no room for further improvement. For other cases with subquestioning, we see an improvement of up to 1 point. Finally, we observe that the *Selection* module is far from perfect and has room for further improvement, as seen from the upper bounds. A *Selection* method ought to look at features of the two answers that turn out to be correlated with correctness, and we hypothesize that models fine-tuned specifically for *Selection* may prove more effective than few-shot learning at identifying these features.

The right half of Tab. 2 is the cheaper method, because we observe $\psi_{ask}$ resamples on only 5-15% of the examples rather than all of them. A tradeoff between accuracy and cost is shown in Fig. 4.

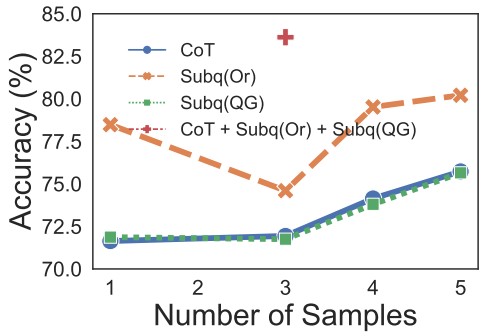 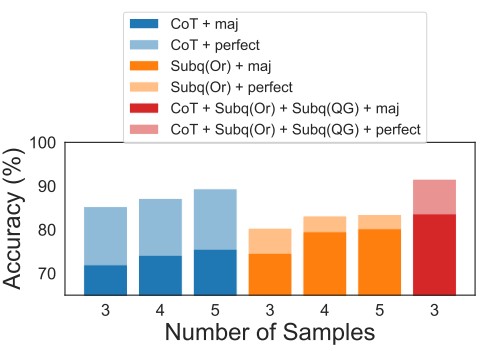

(a) Impact of the number of samples on accuracy when majority voting is used for *selection*.

(b) Comparison of perfect *selection* ("+ perfect") vs. majority voting ("+ maj") across different strategies.

Figure 3: The + in graph (a) shows that majority voting with 3 diverse samples (CoT + Subq(Or) + Subq(QG)) outperforms both CoT and Subq(Or) even with 5 samples. Graph (b) shows the potential of the *selection* method when a perfect selector is used. It can be thought of as the upper bound of the selection mechanism. Both figures are for the GSM8K dataset.

**Selection and Voting** Unweighted majority vote has been one of the most popular *Selection* methods in past work (Wang et al., 2022; Lewkowycz et al., 2022; Zheng et al., 2023), since it requires no training. The two lines in Fig. 3(a) generally show improvement from *Sampling* more times from the same model (at temperature 0.7) and *Selecting* by majority vote.

Recalling that the left half of Tab. 2 showed benefit from ensembling independent samples from 2 different *Sampling* methods (up to 81.34 accuracy when oracle subquestions are allowed), we observe that majority vote is a convenient way to do so for 3 different methods (where all methods can now use temperature 0). This achieves 83.62 accuracy, as shown by the ⋆ in Fig. 3(a). Of course, model-based *Selection* could potentially do even better than majority voting. The 7 points for $k \geq 3$ in (a) are repeated as the dark bars in Fig. 3(b), with the light bars showing the upper bounds that could be achieved by replacing majority voting with a perfect *Selection* method. The best upper bound corresponds again to the use of 3 different methods. In principle, one could ensemble over a larger set by allowing each of the 3 methods to contribute multiple samples.

## 5.2 STRATEGYQA

**Simply resampling does not discover the unknown → A need for tools** For this question answering task, we observe in Tab. 3 that accuracy is harmed by *Conditional Resampling* with the same *Sampling* method, without *Selection*, as was sometimes the case for GSM8K. Here, however, *Selection* usually does not repair the problem, perhaps because StrategyQA requires factual knowledge. When the model lacks the necessary knowledge, Self-Ask will be insufficient. A real example at the bottom of Fig. 5 shows how resampling can preserve an incorrect model-generated claim.

To help the model decide whether and how to revise the answer, we try including relevant facts (provided by StrategyQA) into the resampling prompt to simulate the result one may get by using an external tool like a fact retriever. As Tab. 3 shows, this yields a 2-point improvement ("Facts$_{re}$" vs. "Internal$_s$") over *Sampling*, for both CoT and Subq (QG).

We assume that tool invocations are expensive, which is why we include facts only during *Conditional Resampling*. In practice, the initial result is revised only 10–35% of the time, and therefore "Facts" does not need to invoke a tool call for every input example.[5] To achieve this speedup, we do not include facts in the prompt when initially calling to $\psi_{ask}$ to decide whether to resample, but only when we actually generate $y_{next}$.

---

[5] However, if the facts were included during *Sampling*, the performance can increase beyond 90%.

Table 3: Comparing different strategies for the StrategyQA (top) and Auto Debugging (bottom) datasets. For StrategyQA, external facts are provided to the model ("Facts") versus relying on the model's internal capabilities ("Internal"). The upper bound from using a *Selection* oracle is given in square brackets. Subscripts "$_s$" and "$_{re}$" refer to *Sampling* and *Resampling* respectively.

| Method | Sampling | Conditional Resampling | | Selection | |
|---|---|---|---|---|---|
| Knowledge Source: | Internal$_s$ | Internal$_{re}$ | Facts$_{re}$ | Int$_s$ vs. Int$_{re}$ | Int$_s$ vs. Facts$_{re}$ |
| **StrategyQA** | | | | | |
| CoT | 77.18 | 74.54 | **79.02** | 75.76 | 78.41 |
| Subq (Or) | 85.91 | 78.97 | 84.69 | 85.30 | **86.30** |
| Subq (QG) | 78.16 | 74.69 | **80.40** | 78.78 | 80.00 |
| **Auto Debugging** | | | | | |
| Answer Only | 73.52 | 82.35 | | 88.23 [91.20] | |
| CoT | 70.58 | 73.52 | | 73.52 [73.52] | |
| Answer Only → CoT | - | - | | 85.29 [88.23] | |

## 5.3 CODE DEBUGGING

**The effectiveness of SCREWS** For the code analysis task, we observed that the Answer Only method achieves similar scores to CoT,[6] as reported in the bottom half of Tab. 3, suggesting that no particular *Sampling* method is superior on all datasets. However, we see the benefits of using SCREWS, as we find that with Answer Only, adding *Conditional Resampling* followed by *Selection* leads to a performance boost of 15 points (from 73.52 to 88.23). While the dataset size limits our ability to make concrete conclusions, the findings here support the conclusions drawn on other datasets: *Resampling* and *Selection* lead to benefits and heterogenous sampling can prove effective.

## 6 ADDITIONAL ANALYSIS

**Total Cost** SCREWS supports many methods with different cost/accuracy tradeoffs. Fig. 4 displays the strategies that use CoT and Subq (QG) on GSM8K. The cost is represented as the total count of input tokens (prompt + query) and output tokens for all LLM calls needed by that strategy, averaged over test examples. Generally, Subq (QG) is expensive as it is costly to call $\psi_{\text{question}}$. However, it is affordable to use it in *Conditional Resampling* only (■), since resampling only occurs 10–15% of the time. This method is both cheaper and more accurate than *Sampling* either with Subq (QG) (+) or 3 times with CoT (•). Appendix A discusses a detailed breakdown of each module's input and output token costs.

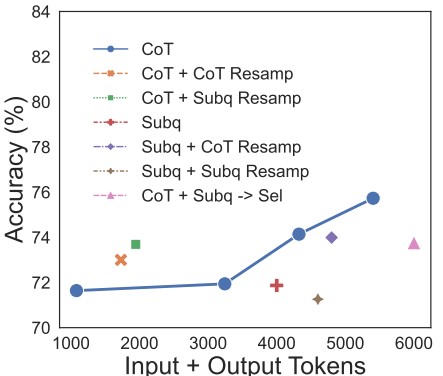

Figure 4: On GSM8K, sampling cost vs. accuracy. The blue line (copied from Fig. 3(a)) shows a baseline of majority voting over $k \in \{1, 3, 4, 5\}$ CoT samples. The shaped points are the other strategies from Sec. 5.1 that use CoT and Subq (QG).

**More Revision Steps** Sec. 5.1 showed that on GSM8K, *Sampling* with Subq (Or) (78.62 accuracy) is improved slightly by *Conditional Resampling* with CoT (78.99) and then *Selection* (79.22). Like Madaan et al. (2023), we did not find much benefit from additional iterations of *Conditional Resampling*+*Selection*: a second iteration gives 79.45, and a third gives 79.52. These small improvements probably do not justify the added cost.

**Larger LLMs** Replacing ChatGPT with GPT-4 (OpenAI, 2023) greatly increased the *Sampling* accuracy on GSM8K, to 91.45 for CoT and 90.80 for Subq (Or). Choosing between those two sam-

---

[6]We do not experiment with subquestion decomposition as subquestions are not part of this dataset.

ples with GPT-4-based *Selection* further increased the accuracy to 93.10, which falls between the accuracy of majority voting over $k$=3 and $k$=4 CoT samples from GPT-4 (92.94 and 93.93 respectively). Even using ChatGPT-based *Selection* achieved 92.58, which improves over CoT alone.

**Selected Examples**   The top two examples of Fig. 5, on the GSM8K dataset, demonstrate the effectiveness of the *Selection* module. The first example shows how an error introduced by *Conditional Resampling* can be reverted by *Selection*. The second example shows how a correction found by *Conditional Resampling* can be kept by *Selection*. The last example in Fig. 5, on the StrategyQA dataset, illustrates that ordinary *Resampling* is unlikely to correct an incorrect fact generated by the LLM. However, providing the correct facts during *Resampling* gives the model access to new information, leading to the correct answer.

---

**Question** Gunter is trying to count the jelly beans in a jar. He asks his friends how many they think are in the jar. One says 80. Another says 20 more than half the first one. A third says 25% more than the first one. What is their average guess?

**Sampling** (A) The second person guesses $20 + 1/2 * 80 = $ <<20 + 1/2 * 80 = 60>>60 jelly beans. The third person guesses $80 * 1.25 = $ <<80 * 1.25 = 100>>100 jelly beans. The total number of jelly beans is $80 + 60 + 100 = $ <<80 + 60 + 100 = 240>>240. The average guess is $240/3 = $ <<240/3 = 80>>80. The answer is 80.✓

**Conditional Resampling** (B) The average guess is $(80+60+1.25*80)/3 = $ <<(80+60+1.25*80)/3 = 85>>85 jelly beans. The second person guesses $80/2 + 20 = $ <<80/2 + 20 = 60>>60 jelly beans. The average guess is $(80 + 60 + 1.25 * 80)/3 = $ <<(80 + 60 + 1.25 * 80)/3 = 85>>85 jelly beans. The answer is 85. ✗

**Selection** (A) ✓

---

**Question** Adam has \$100 and wants to spend it to open a rock stand. He can buy rocks for \$5 each and sell them for \$7 each. If he invests all his money in the rock stand but only sells 60% of his inventory, how much money does he lose?

**Sampling** (A) Adam can buy \$100/\$5 per rock = <<100/5 = 20>>20 rocks. Adam sells 60/100 x 20 rocks = <<60/100 * 20 = 12>>12 rocks. Adam makes \$7 x 12 rocks = \$<<7 * 12 = 84>>84. Adam loses \$100 - \$5 x 20 rocks = \$<<100 − 5 * 20 = 0>>0. He invested all his money and sold all the rocks he bought, so he didn't lose any money. The answer is 0. ✗

**Conditional Resampling** (B) Adam can buy \$100/\$5 per rock = <<100/5 = 20>>20 rocks. Adam sells 60/100 x 20 rocks = <<60/100*20 = 12>>12 rocks. Adam makes \$7 x 12 rocks = \$<<7 * 12 = 84>>84. However, he spent \$5 x 20 rocks = \$<<5 * 20 = 100>>100, so he loses \$100 - \$84 = \$<<100 − 84 = 16>>16. The answer is 16. ✓

**Selection** (B) ✓

---

**Question**: Would Alexander Hamilton have known about koalas?

**Sample**: Alexander Hamilton died in 1804. The first koala specimen was collected in 1792. It is possible that Hamilton knew about koalas, but it is unlikely. The answer is Unknown ✗

**Conditional Resampling**: Alexander Hamilton died in 1804. The first koala specimen was collected in 1792. It is possible that Hamilton knew about koalas. The answer is True ✗

**Facts**: *Alexander Hamilton died in 1804. The first published depiction of a koala was in 1810*

**Tool-based Resample**: No, Alexander Hamilton would not have known about koalas. The answer is False ✓

---

Figure 5: The top two examples demonstrate the importance of the *Selection* module for the GSM8K dataset. The last example shows how tool use ("Facts") can be helpful for the StrategyQA dataset.

## 7   CONCLUSION AND FUTURE WORK

We have proposed SCREWS, a modular reasoning-with-revisions framework to answer reasoning questions with LLMs. Based on our experiments we conclude the following: 1) **Selection plays an important role**: Although *Conditional Resampling* often improves the result of *Sampling*, *Selection* can help avoid errors from the case where it does not. It was beneficial on all three datasets; 2) **Heterogeneous vs. homogeneous resampling**: Using different reasoning methods for *Sampling* and *Conditional Resampling* can lead to higher accuracy, with or without *Selection*; 3) **Missing external knowledge hurts *Conditional Resampling***: Resampling cannot fix incorrect facts generated by the model. Tool-based resampling can therefore get better results (as simulated using StrategyQA); and 4) **No uniformly best strategy**: There was no clear winning method for each of the modules. Simple baseline methods sometimes beat more complex ones: CoT uses only one call to $\psi$ and beats Subq (QG) in GSM8K, always selecting $y_{next}$ beats self-select for StrategyQA with "Facts," and Answer Only works surprisingly well for Code Debugging.

SCREWS combines the three important modules *Sampling*, *Conditional Resampling* and *Selection* in a modular framework. However, the best configuration of modules may vary by task and could be identified through a method such as exhaustive search, Monte Carlo Tree Search, or reinforcement learning. The modules themselves could be fine-tuned to improve end-to-end performance. We leave this for future work.

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

## A  TOKEN COST

Tab. A shows the token cost of input and output for each module in SCREWS. Due to its iterative nature, subquestion decomposition requires on average four times more input tokens than the other modules. For *Conditional Resampling*, the model first predicts whether it wants to modify its output or not, using one token ("Yes" or "No") for each sample and then only for the answers starting with "No", it resamples. For the *Selection* module, the model chooses one of the two samples presented to it, using one token (A or B) for the output.

| Method | Input Tokens | Output Tokens | Total Tokens |
|---|---|---|---|
| **Subquestion generation step $\psi_{\text{question}}$** | | | |
| Subq (QG) | 360 | 180 | 540 |
| **Sampling step $\psi$** | | | |
| CoT | 774 | 307 | 1081 |
| CoT ($k = 5$) | 3870 | 1530 | 5400 |
| Subq (Or) | 3187 | 413 | 3600 |
| Subq (QG) | 3121 | 434 | 3555 |
| **Conditional Resampling step $\psi_{\text{ask}}$** | | | |
| CoT | 869 | 105 | 1184 |
| Subq (Or) | 3525 | 131 | 3656 |
| Subq (QG) | 3780 | 136 | 3916 |
| **Selection step $\psi_{\text{select}}$** | | | |
| Selection | 1296 | 1 | 1297 |

Table 4: Average counts of input and output tokens for each choice of each module (step) in SCREWS. Many of the methods in Tab. 1 need to call multiple modules. We remark that the input tokens at each step include output tokens from previous steps. The counts shown for later steps average not only over examples, but also over choices of method for the previous steps.

## B  PROMPTS

Below are abbreviated versions of the prompts used in the experiments, including instructions and demonstrations. For readability, we show only 1–2 demonstrations in each prompt. In each demonstration, the demonstrated result string is highlighted for the reader's convenience, but this highlighting is not included in the prompt. Each prompt shown would be followed by the test question and then the cue (e.g., "Answer:") that indicates that a result string should follow.

### B.1  SAMPLING

For Chain of Thought (CoT) and Subquestion Decomposition for GSM8K and StrategyQA, 5-shot prompts were used. For Auto Debugging, a 1-shot prompt was used.

#### B.1.1  CHAIN OF THOUGHT

**GSM8K**
I am a highly intelligent question answering bot. I will answer the last question 'Question' providing equation in $<<>>$ format in step by step manner.

Question: James writes a 3-page letter to 2 different friends twice a week. How many pages does he write a year?
Answer: He writes each friend $3 * 2 = <<3 * 2 = 6>>6$ pages a week. So he writes $6 * 2 = <<6 * 2 = 12>>12$ pages every week. That means he writes $12 * 52 = <<12 * 52 = 624>>624$ pages a year. The answer is 624

**StrategyQA**

You are a highly intelligent question answering bot. You will answer the question 'Question' in as details as possible.

Question: Is coal needed to practice parachuting?

Answer: Parachuting requires a parachute. Parachutes are made from nylon. Nylon is made from coal. The answer is True

**Auto Debugging**

Answer the 'Question' based on the provided code and provide explanation.

Question:

```python
def f1():
    return str(x) + 'hello'
def f2():
    return f1(2*x)
x = f2(524)
```

What is the value of x at the end of this program?

Output: First, x = 2 * 524 = 1048 and then 'hello' is appended to it. So x becomes 1048hello

### B.1.2   SUBQUESTION DECOMPOSITION

While subquestion decomposition uses a single prompt, each example requires multiple API calls because the next subquestion needs to be appended to the prompt.

**GSM8K**

I am a highly intelligent question answering bot. I will answer the last question 'Q' providing equation in << >> format keeping the Problem and previous Q and A into account.

Problem: There are 5 houses on a street, and each of the first four houses has 3 gnomes in the garden. If there are a total of 20 gnomes on the street, how many gnomes does the fifth house have?

Q: How many gnomes are in the first four houses?

A: In the first four houses, there are a total of 4 houses * 3 gnomes = $<<4*3 = 12>>$12 gnomes. The answer is 12

Q: How many gnomes does the fifth house have?

A: Therefore, the fifth house had 20 total gnomes - 12 gnomes = $<<20 - 12 = 8>>$8 gnomes. The answer is 8

**StrategyQA**

You are a highly intelligent question answering bot. You will answer the last question 'Q' keeping the Problem and previous Q and A into account and then answer the Final Question based on all the previous answer 'A'.

Problem: Is coal needed to practice parachuting?

Q: What is one of the most important item that you need to go parachuting?

A: Parachuting requires a parachute.

Q: What is #1 made out of?

A: Parachutes are made from nylon.

Q: Is #2 originally made from coal?

A: Nylon is made from coal.

Final Question: Is coal needed to practice parachuting?

Final Answer: True

### B.1.3   ANSWER ONLY

Answer Only was only used for Auto Debugging in a 1-shot manner and that one example is provided below:

Input:

```python
def f1():
    return str(x) + 'hello'
```

```
def f2():
    return f1(2*x)
x = f2(524)
```

What is the value of x at the end of this program?
Output: `1048hello`

---

## B.2 CONDITIONAL RESAMPLING

We used 4-shot prompting for *conditional resampling* (two positive and two negative) for GSM8K and StrategyQA and 2-shot (one positive and one negative) for Auto Debugging. One pair of each is presented below:

### B.2.1 CHAIN OF THOUGHT

**GSM8K**
Question: Natalia sold clips to 48 of her friends in April, and then she sold half as many clips in May. How many clips did Natalia sell altogether in April and May?
Answer: In April, Natalia sold 48 clips. In May, Natalia sold half of the toal clips meaning $72/2<<72/2 = 36>>36$ clips. So, Natalia sold a total of $48 + 36<<48 + 36 = 84>>84$ clips in April and May. The answer is 84
You are a math teacher. Do you think the reasoning process for the given problem is correct? Let's check the 'Answer' in details, and then decide 'Yes' or 'No' and then write the correct 'Final Answer'.
Answer: No, the reasoning is incorrect. Final Answer: Natalia sold $48/2 = <<48/2 = 24>>24$ clips in May. So she sold a total of $48 + 24<<48 + 24 = 72>>72$ clips. The answer is 72

Question: There are 5 houses on a street, and each of the first four houses has 3 gnomes in the garden. If there are a total of 20 gnomes on the street, how many gnomes does the fifth house have?
Answer:In the first four houses, there are a total of 4 houses * 3 gnomes $= <<4 * 3 = 12>>12$ gnomes. Therefore, the fifth house had 20 total gnomes - 12 gnomes $= <<20 - 12 = 8>>8$ gnomes. The answer is 8
You are a math teacher. Do you think the reasoning process for the given problem is correct? Let's check the 'Answer' in details, and then decide 'Yes' or 'No' and then write the correct 'Final Answer'.
Answer: Yes, the reasoning is correct. Final Answer: In the first four houses, there are a total of 4 houses * 3 gnomes $= <<4 * 3 = 12>>12$ gnomes. Therefore, the fifth house had 20 total gnomes - 12 gnomes $= <<20 - 12 = 8>>8$ gnomes. The answer is 8

---

**StrategyQA**
Question: Could Durian cause someone's stomach to feel unwell?
Answer: Durian has a pungent odor that many people describe as being similar to feet and onions. Unpleasant smells can make people feel excited and they like it. The answer is False
You are an expert teacher. Do you think the reasoning process for the given problem is correct? Let's check the 'Answer' in details, and then decide 'Yes' or 'No' and then write the correct 'Final Answer'.
Answer: No, the reasoning is incorrect. Final Answer: Durian has a pungent odor that many people describe as being similar to feet and onions. Unpleasant smells can make people feel nauseous. The answer is True

Question: Was Daniel thrown into the lion's den in the New Testament?
Answer:The Book of Daniel is a book in the Old Testament of the Bible. The Bible is divided into the Old Testament and the New Testament. The New Testament focuses on four Gospels regarding the life of Jesus. The answer is False
You are an expert teacher. Do you think the reasoning process for the given problem is correct? Let's check the 'Answer' in details, and then decide 'Yes' or 'No' and then write the correct 'Final Answer'.
Answer: Yes, the reasoning is correct. Final Answer: The Book of Daniel is a book in the Old Testament of the Bible. The Bible is divided into the Old Testament and the New Testament. The New Testament focuses on four Gospels regarding the life of Jesus. The answer is False

---

**StrategyQA (Resampling with facts)**
You are a highly intelligent question answering bot. You will answer the question 'Question' in as details as possible. 'Facts' are provided to assist you in answering the questions.
Question: Are vinegar pickled cucumbers rich in lactobacillus?
Facts: Pickles made with vinegar are not probiotic and are simply preserved. Pickles made through a soak in a

salt brine solution begin to ferment because of lactobacillus.
Answer: No, vinegar does not contain lactobacillus. The answer is False

Question: Does Masaharu Morimoto rely on glutamic acid?
Facts: Masaharu Morimoto is a Japanese chef. Japanese cuisine relies on several forms of seaweed as ingredients and flavorings for broth like kombu dashi. Glutamic acid has been identified as the flavoring component in kombu seaweed.
Answer: Yes, Japanese chef uses a lot of glutamic acid. The answer is True

**Auto Debugging**
Input:

```python
def f1():
    return str(x) + 'hello'
def f2():
    return f1(2*x)
x = f2(524)
```

What is the value of x at the end of this program?
Output: 1048hello
Verdict: Yes, the answer is correct.
Final Answer: 1048hello

Input:

```python
def f1():
    return str(x) + 'hello'
def f2():
    return f1(2*x)
x = f2(524)
```

What is the value of x at the end of this program?
Output: 524
Verdict: No, the answer is incorrect.
Final Answer: 1048hello

### B.2.2 SUBQUESTION DECOMPOSITION

**GSM8K**
For each subquestion, the main problem and all previous subquestions along with the model-generated solutions are provided in order to solve the current subquestion.

Here is a math question and its solution.
Problem: Noah is a painter. He paints pictures and sells them at the park. He charges \$60 for a large painting and \$30 for a small painting. Last month he sold eight large paintings and four small paintings. If he sold twice as much this month, how much is his sales for this month?
How much did Noah earn from the large paintings? Noah earned \$60/large painting x 8 large paintings = \$<<60 * 8 = 480>>480 for the large paintings. The answer is 480
Question: How much did Noah earn from the small paintings?
Answer: He also earned \$60/small painting x 4 small paintings = \$<<60 * 4 = 240>>240 for the small paintings. The answer is 240
You are a math teacher. Do you think the reasoning process for the given problem is correct? Let's check the 'Answer' in details, and then decide 'Yes' or 'No' and then write the correct 'Final Answer'.
Answer: No, the reasoning is incorrect. Final Answer: He also earned \$30/small painting x 4 small paintings = \$<<30 * 4 = 120>>120 for the small paintings. The answer is 120

Here is a math question and its solution.
Problem: To make pizza, together with other ingredients, Kimber needs 10 cups of water, 16 cups of flour, and 1/2 times as many teaspoons of salt as the number of cups of flour. Calculate the combined total number of cups of water, flour, and teaspoons of salt that she needs to make the pizza.
How many teaspoons of salt does Kimber need? To make the pizza, Kimber half as many teaspoons of salt as

the number of cups of flour, meaning she needs 1/2*16 = $<<16*1/2 = 8>>$8 teaspoons of salt. The answer is 8

How many cups of flour and teaspoons of salt does Kimber need? The total number of cups of flour and teaspoons of salt she needs is 8+16 = $<<8 + 16 = 24>>$24. The answer is 24

Question: How many cups of water, flour, and salt does Kimber need?

Answer: She also needs 10 cups of water, which means the total number of cups of water and flour and teaspoons of salt she needs is $24 + 10 = <<24 + 10 = 34>>$34. The answer is 34

You are a math teacher. Do you think the reasoning process for the given problem is correct? Let's check the 'Answer' in details, and then decide 'Yes' or 'No' and then write the correct 'Final Answer'.

Answer: ==Yes, the reasoning is correct. Final Answer: She also needs 10 cups of water, which means the total number of cups of water and flour and teaspoons of salt she needs is $24 + 10 = <<24 + 10 = 34 > >>$34. The answer is 34==

---

### StrategyQA

Here is a question and its answer.
Context: Would a diet of ice eventually kill a person?
Ice is the solid state of what? Ice can be melted into water, which consists of hydrogen and oxygen.
What nutrients are needed to sustain human life? Humans need carbohydrates, proteins, and fats that are contained in foods.
Question: Are most of #2 absent from #1?
Answer: Water does not contain fat, carbohydrates or protein.
You are an expert teacher. Based on the provided context, do you think the reasoning process for the given problem is correct? Let's check the 'Answer' in details, and then decide 'Yes' or 'No' and then write the correct 'Final Answer'.
Answer: ==Yes, the reasoning is correct. Final Answer: Water does not contain fat, carbohydrates or protein.==

Here is a question and its answer.
Context: Can binary numbers and standard alphabet satisfy criteria for a strong password?
Which characters make up binary numbers? Binary numbers only contain 0 and 1.
Which characters make up the standard English alphabet? The standard alphabet contains twenty six letters but no special characters.
Question: Does #1 or #2 include special characters or symbols?
Answer: Yes, it contains all the special characters.
You are an expert teacher. Based on the provided context, do you think the reasoning process for the given problem is correct? Let's check the 'Answer' in details, and then decide 'Yes' or 'No' and then write the correct 'Final Answer'.
Answer: ==No, the reasoning is incorrect. Final Answer: Neither binary digits nor English alphabets consists of any special characters which is needed for a strong password.==

---

## B.3 SELECTION

The LLM-based selection module $\psi_{\text{select}}$ uses a 2-shot prompt. The 2 demonstrations in the prompt are shown below, for each dataset.

### GSM8K

You are an expert math teacher. You are provided with a question and two answers. Lets check the 'Answer choices' step by step, and then decide which answer is correct '(A)' or '(B)'
Question: Natalia sold clips to 48 of her friends in April, and then she sold half as many clips in May. How many clips did Natalia sell altogether in April and May?
Answer choices:
(A) In April, Natalia sold 48 clips. In May, Natalia sold 24 clips. So, Natalia sold a total of 72 clips in April and May. The answer is 72. So in May she sold 48 clips. Total clips sold in April and May = $72 + 48 = <<72 + 48 = 120>>$120. The answer is 120
(B) Natalia sold $48/2 = <<48/2 = 24>>$24 clips in May. The answer is 24. Natalia sold $48 + 24 = <<48 + 24 = 72>>$ clips altogether. The answer is 72
Answer: ==(B)==

You are an expert math teacher. You are provided with a question and two answers. Lets check the 'Answer choices' step by step, and then decide which answer is correct '(A)' or '(B)'

Question: Dolly has two books. Pandora has one. If both Dolly and Pandora read each others' books as well as their own, how many books will they collectively read by the end?
Answer choices:
(A) There are a total of $2 + 1 = <<2 + 1 = 3>>3$ books. The answer is 3. Dolly and Pandora both read all 3 books, so 3 books/person x 2 people = $<<3 * 2 = 6>>6$ books total. The answer is 6
(B) The total number of books are $2 * 1 = <<2 * 1 = 2>>2$ books. The answer is 2. Dolly and Pandora read each other's books as well as their own, so the total number of books they read is 3 books. The answer is 3
Answer: (A)

---

### StrategyQA

You are the expert in the field. You are provided with a question and two answers. Lets check the reasoning process of each of the answer step by step, and then decide which answer is correct '(A)' or '(B)'
Question: Could Durian cause someone's stomach to feel unwell?
Answer choices:
(A) Durian has a pungent odor that many people describe as being similar to feet and onions. Unpleasant smells can make people feel nauseous. The answer is True
(B) Durian has a pungent odor that many people describe as being similar to feet and onions. Unpleasant smells can make people feel excited and they like it. The answer is False
Answer: (A)

You are the expert in the field. You are provided with a question and two answers. Lets check the reasoning process of each of the answer step by step, and then decide which answer is correct '(A)' or '(B)'
Question: Was Daniel thrown into the lion's den in the New Testament?
Answer choices:
(A) The Book of Daniel is a book in the New Testament of the Bible. The Bible is divided into the Old Testament and the New Testament. The New Testament focuses on the life of Daniel. The answer is True
(B) The Book of Daniel is a book in the Old Testament of the Bible. The Bible is divided into the Old Testament and the New Testament. The New Testament focuses on four Gospels regarding the life of Jesus. The answer is False
Answer: (B)

---

### Auto Debugging

You are an expert Python debugger. You are provided with a question and two answers. Your job is to decide which answer is correct '(A)' or '(B)'
Question:

```python
def f1():
    return str(x) + 'hello'
def f2():
    return f1(2*x)
x = f2(524)
```

What is the value of x at the end of this program?
Answer choices:
(A) `524hello`
(B) `1048hello`
Answer: (B)

---

## B.4 Question Generation

5-shot prompts were used for generating subquestions for GSM8K dataset. An example is provided below:

### GSM8K

I am a highly intelligent question generation bot. I will take the given question 'Q' and will decompose the main question into all 'subquestions' required to solve the question step by step.

Q: James writes a 3-page letter to 2 different friends twice a week. How many pages does he write a year?
Subquestions: How many pages does he write each week? How many pages does he write every week? How many pages does he write a year?

## StrategyQA
I am a highly intelligent question generation bot. I will take the given question 'Q' and will decompose the main question into all 'subquestions' required to solve the question step by step.

Q: Can you buy Casio products at Petco?
Subquestions: What kind of products does Casio manufacture? What kind of products does Petco sell? Does #1 overlap with #2?

