# OpenReview forum: "SCREWS: A Modular Framework for Reasoning with Revisions"
_ICLR.cc/2024/Conference — ICLR 2024 Conference Withdrawn Submission_

### Official Review · Reviewer_4Ly4 · 2023-10-21

**Soundness:** 2 fair
**Presentation:** 2 fair
**Contribution:** 2 fair
**Rating:** 3
**Confidence:** 4

**Summary:**

This paper presents SCREWS, a methodology for reasoning with revisions. The pipeline includes three stages: sampling, conditional resampling, and selection. The experiments demonstrated that using different strategies for sampling and conditional resampling can boost the reasoning performance of the GPT-3.5 language model.

**Strengths:**

The proposed framework is general and modular, meaning various techniques can be employed in different stages of it.

**Weaknesses:**

1. > A student preparing for an exam may use deductive reasoning to solve problems and inductive reasoning to verify the results

    This is surely the wrong way around?

2. Figure 2 is too visually complicated to be helpful. It's better to present a simplified and more abstract pipeline than listing every component.

3. This is the main thing I am unsure about: In tables 1 and 2, the results are supposed to demonstrate the usefulness of the resampling strategy. However, in table 1, only 4 out of 9 pairings are statistically significant. Also, when using Subq (Or) as the sampling strategy, it does not seem to matter much (or have statistical significance) as to which conditional resampling strategy is used. Does this maybe suggest that there is a saturation, or utility limitation of SCREWS, if the sampling strategy is good enough?

    In table 2, although conditional sampling is cheaper, independent sampling does have significantly higher performances (and upper bounds given oracle selectors). The figure 4 provides further breakdown of the accuracy - cost relation and no strategy really beats CoT in absolute performance. This leaves the question of the overall usefulness of SCREWS uncertain.

**Questions:**

Could you provide ablation studies to show clearly that given the same computational cost, SCREWS can perform better than naive CoT? If so, I can be convinced of its effectiveness.

---

> ### Author Response · Authors · 2023-11-15
> **Usefulness of SCREWS**
>
> We thank the reviewer for taking the time to review our work.
>
> We can simplify the main figure by making it clearer and put it in the appendix. Thanks for the suggestion.
>
> As mentioned above, refinement does not work well for reasoning tasks and often leads to worse results (Self-Refine (https://arxiv.org/abs/2303.17651) and recent work (https://arxiv.org/abs/2310.01798) ).
> In our case, we improve the results with our novel component of heterogeneous resampling and selection. Although we have not reached the upper bounds, we have shown the usefulness of the framework to try different strategies, and it can be improved with more such combinations.
>
> Subquestion decomposition is an expensive strategy compared to CoT because we have to sample iteratively for each decomosed questions. Deciding when to resample and resampling only selected samples is a cost-effective strategy compared to resampling all samples, and thus the method is cost effective. We have shown in Fig. 3a that CoT with 5 samples can achieve an accuracy of 75, while our strategy with 3 samples can achieve 83. This is the effectiveness of the SCREWS strategy. We can add more such examples in the paper.

---

### Official Review · Reviewer_Na8j · 2023-10-30

**Soundness:** 3 good
**Presentation:** 3 good
**Contribution:** 3 good
**Rating:** 6
**Confidence:** 3

**Summary:**

This paper studies refinement and revision in reasoning. They propose a modular framework for improving reasoning with revisions. The proposed framework unifies several previous approaches under a common framework but also reveals several novel strategies for identifying improved reasoning chains. It consists of three main modules, Sampling, Conditional Resampling, and Selection, each consisting of sub-modules that can be hand-selected per task. The framework is then implemented with GPT-3.5-turbo and GPT-4 and is evaluated on multiple benchmarks for arithmetic reasoning, multi-hop question answering, and code analysis. The proposed strategies achieve substantial improvements over vanilla strategies. The heterogeneous sampling strategy is demonstrated useful in the experiments. They also discuss the importance of a model-based selection strategy.

**Strengths:**

This paper studies the problem of revisions in reasoning, including reducing errors introduced by revision and alleviating homogenous revisions, which are important research questions for current large language model reasoning. The authors propose a unified framework to address the questions. Many previous works can be viewed as an instance of the proposed framework. As a result, the framework is convenient for ablating the strategies during the pipeline. The experiments and analyses are comprehensive. The proposed strategies are effective. And the experimental findings are inspiring.

**Weaknesses:**

Please see the questions listed below.

**Questions:**

Q1: How do you choose the specific sub-modules (e.g., self-ask/tool use for conditional resampling, LLM-based selection/Rule-based selection for selection) for each of the three modules in the framework?

---

> ### Author Response · Authors · 2023-11-15
> **Extension of SCREWS modules/sub-modules**
>
> We thank the reviewer for taking the time to review our work.
>
> This is an important question, and with SCREWS we are not limited to any number of submodules. We have only demonstrated those that have been used in previous work combined with our proposed new sub-modules (for example, majority voting is a commonly used sub-module for selection and we extend it to LLMs making decision by proposing self-select, ...).
> However, our framework can be easily extended to any new modules/sub-modules.

---

### Official Review · Reviewer_toqi · 2023-11-01

**Soundness:** 2 fair
**Presentation:** 3 good
**Contribution:** 2 fair
**Rating:** 3
**Confidence:** 5

**Summary:**

This paper proposes a framework, called SCREWS, with modular components for reasoning tasks where revisions and selection are needed. The framework contains three modules, Sampling, Conditional Resampling, and Selection. Each of the modules can then be implemented with several alternatives. The authors conducted experiments on GSM8K, StrategyQA, and Auto Debugging, using gpt-3.5-turbo, aiming to see how different combinations of modules can affect the task performance. The important observations include: 1) conditional resampling helps when it is based on a different method than the sampling, 2) a good selection is promising for improving the task performance, but the current selection method still falls short in it, and 3) enabling tools is critical for StrategyQA where additional facts are beneficial.

**Strengths:**

1. The paper has touched upon a popular topic of LLM reasoning, especially when iterative revisions are needed. The proposed framework summarized the typical implementation of different modules.
2. The paper conducted experiments with different combinations of module instantiations and investigated their effectiveness. The experimental results have led to several interesting takeaway messages.
3. The paper is easy to follow.

**Weaknesses:**

The contribution of this paper seems to be incremental, as it is mainly an empirical exploration of existing module implementations. While the experimental results led to interesting observations, these observations are mostly expected, whereas the more critical questions, such as how to improve the existing selection method, are not well addressed.

**Questions:**

I found the tool use experiment of StrategyQA a bit confusing.
1. I wonder if the conditional resampling can still be helpful if the LLM is configured to access the retrieved fact in its initial sampling?
2. The setup seems to directly provide relevant facts to the conditional resampler, and the model does not actually use any Web search tool for fact retrieval. Is this the major reason for task improvement? I wonder in the more realistic case of using an external tool, if the same improvement can be observed (considering the potential noise, long retrieved passages, etc.).

---

> ### Author Response · Authors · 2023-11-15
> **Why refinement is not so simple for reasoning**
>
> We thank the reviewer for taking the time to review our work.
>
> As mentioned in response to 2Dnf, the paper proposed heterogeneous sampling as an approach to refinement, and later a choice between the refined answer and the original prediction as an option to "roll back" in case of errors in refinement. These two components are missing in all previous work. The proposed framework allows different previous strategies to mix and match different components to get the best refinement strategy for a given task. The framework is one of the contributions, not just the contribution.
>
> Moreover, expecting improvement through refinement is a common misconception, and it often leads to worse results for reasoning tasks (Self-Refine (https://arxiv.org/abs/2303.17651) and recent work (https://arxiv.org/abs/2310.01798) ).
>
> If the LLM already knows the facts, it will not make factual errors (we mention this in the paper and for strategyQA, providing facts in the first steps leads to 90% accuracy, higher than refinement). But we are testing if LLMs can judge their mistakes, and to avoid factual errors again, we provide the factual info.
> Testing this with any other tool was out of scope, as we need to disambiguate the errors due to tools vs. LLMs errors.

---

### Official Review · Reviewer_2Dnf · 2023-11-06

**Soundness:** 2 fair
**Presentation:** 3 good
**Contribution:** 2 fair
**Rating:** 3
**Confidence:** 4

**Summary:**

SCREWS, a modular framework for reasoning with revisions.  The author observe that these revisions can introduce errors, in which case it is better to roll back to a previous result. So there should be a framework that decide we should accept the current revision or not.

The proposed approach consists of three steps:
[1] Sampling instantiate SCREWS by fixing the submodules for each module
[2] Conditional Resampling, which decides whether to generate a revision conditioned on the initial sample, and does so if needed.
[3] Selection: all samples and revisions are given to the Selection module, which selects the best one.

Each of the above three modules include several existing effective methods:
Sampling: CoT, decomposition
Condition resampling: Self ask, tool use
Selection:  self-consistency, rule-based etc

**Strengths:**

The paper works on an interesting problem. The paper collects a couple of well known approaches and integrate them into this framework, and provide suggestions on how to use them. The paper objectively reports results, and performs analysis and comparison. Regarding ideas, self-ask with respect to multiple steps of decomposition is quite interesting.

**Weaknesses:**

1 the paper is a collection of existing approaches, the contribution is a bit incremental and the novelty is a bit limited.

2 the effectiveness of the proposed approach is not quite conclusive yet.

- Table 1 the conclusion is sampling and conditional reasamping should use different sampling approach, i.e. CoT + Subq (QG) or Subq (QG) + CoT. However, the improvement is rather incremental (i.e. 73-> 73.99). Especially considering SOTA of GSM8K IS 90+ https://paperswithcode.com/sota/arithmetic-reasoning-on-gsm8k (although we understand the foundation models are different, the effectiveness of the approach is not clear)

- Table 2 “independent sampling” combines Subq (QG) and CoT (74.90) give the best performance than “conditional sampling” (73.99 table 1), which makes me unclear of the effective of conditional reasoning (i.e. combine two samplings are easy and just do majority vote on, i.e. no need to ask LLM whether to resample or not)

- The right half of Tab. 2 shows Selection between the Sampled and Conditionally Resampled. Does that mean the selection module doesn’t bring significant gain?

**Questions:**

See above

---

> ### Author Response · Authors · 2023-11-15
> **Main Contribution**
>
> We thank the reviewer for taking the time to review our work.
>
> 1. The paper proposed heterogeneous sampling as an approach to refinement, and later a choice between the refined answer and the original prediction as an option to "roll back" in case of errors in refinement. These two components are missing in all previous work. The proposed framework allows different previous strategies to mix and match different components to get the best refinement strategy for a given task. The framework is one of the contributions, but we propose ways to improve reasoning during refinement (main contribution).
>
> SoTA is definitely 90+ percent, and we also achieve 93+ accuracy in our work (see the *Larger LLM* part in the *Additional Analysis* section).
>
> 2. Self-Refine (https://arxiv.org/abs/2303.17651) and more recent work (https://arxiv.org/abs/2310.01798) show that reasoning tasks cannot be improved by *direct refinement*, and often lead to worse results. Our component of heterogeneous resampling and selection not only improves refinement scores, but can be done at much lower cost (refinement is not always needed by judging when to refine).
>
> Although independent sampling gives the best performance, it resamples all the time. If we want a low-cost method, we can resample only when needed. But if budget is not a constraint, one should prefer independent sampling. Both are possible with our framework.

---

### Author Response · Authors · 2023-11-15
**Thank you for all the suggestions.**

We would like to thank all reviewers for their useful reviews.
We would like to withdraw the paper to reflect on the suggestions and will incorporate all the feedback in the future submission.
Thank you very much.